# Prediction of Temperature Development of Concrete with Set-Controlling Admixture Based on a New Hydration Kinetics Model

**DOI:** 10.3390/ma16020497

**Published:** 2023-01-04

**Authors:** Yi Yu, Pengfei Zhu, Yanran Shi, Fei Xu, Linhua Jiang, Hongqiang Chu, Ning Xu, Mingwei Liu, Yu Jia, Tao Peng

**Affiliations:** 1Materials & Structural Engineering Department, Nanjing Hydraulic Research Institute, Nanjing 210024, China; 2College of Mechanics and Materials, Hohai University, Nanjing 210098, China; 3College of Civil Engineering, Jiangsu Open University, Nanjing 210019, China

**Keywords:** hydration kinetics, temperature development, set-controlling admixture, cement, concrete

## Abstract

Temperature control is needed in the construction process of massive concrete and it can avoid the concrete cracks. Prediction of temperature development based on a hydration kinetics model can reduce the need for adiabatic temperature rise tests for concrete. However, the existing hydration kinetics model cannot accurately describe the hydration process of cement, thereby limiting the ability to further accurately predict the temperature rise of concrete based on the hydration kinetics model. This paper aims to establish a new hydration kinetics model, which is based on nucleation and growth model, and to predict the temperature development of concrete with set-controlling admixture based on this model. In this paper, the nucleation and growth of hydration products and the diffusion of free water by the modified boundary of nucleation and growth (BNG) model and the modified Fuji and Kondo’s model are described. The relationship between nucleation rate and apparent activation energy and the relationship between effective diffusion coefficient and apparent activation energy are linear. However, the relationship between growth rate and apparent activation is exponential. Finally, the temperature development of concrete can be calculated by the hydration degree of the cement.

## 1. Introduction

Recently, bridge engineering, port engineering and water conservancy engineering have involved massive concrete. After pouring massive concrete, temperature stress will be generated due to the influence of the external temperature and the hydration temperature. If the temperature control is insufficient, it will cause concrete cracking, thus threatening the overall safety of the concrete structure. Therefore, temperature control is needed in the construction process of massive concrete. The prediction of temperature development in concrete is essential to control the temperature of the concrete [1,2].

Hydration kinetics of cement has been modeled in many studies since it is the governing process for the prediction of various properties of concrete. Zhang et al. proposed a hydration kinetics model for predicting the degree of hydration at the initial setting time of cement paste with particle agglomeration [3]. Zhang et al. established a new nucleation and growth model and predicted the hydration heat and microstructure of cement with slag by the new model [4]. Wang et al. modified Tomosawa’s hydration kinetics model and predicted the temperature development of concrete incorporating fly ash/slag by the new model [5]. Therefore, an appropriate hydration kinetics model can effectively predict the properties of concrete and reduce a large number of experiments involving concrete in practical engineering.

Several hydration kinetics models have been developed to determine the hydration degree of cement [6,7,8]. There are still two main types of existing hydration kinetics models: single-particle models and nucleation and growth models. However, the existing single-particle models do not account for particle–particle interaction, and do not accurately represent the overall kinetics of a collection of hydrating particles with a wide range of sizes [9,10]. The existing nucleation and growth models do not account for water to cement and particle size distribution of cement [11,12]. Therefore, a new hydration kinetics model must be established to accurately predict the hydration degree of cement.

With the current diversity application requirements of concrete, various admixtures are widely added to the concrete [13]. Set-controlling admixtures, especially, retarders and accelerators, have a significant influence on the hydration of cement [14,15]. However, the effect of set-controlling admixtures on the hydration kinetics of cement is not clear, which makes it is difficult for exiting hydration kinetics to predict the temperature development of concrete mixed with set-controlling admixtures.

This study aimed to establish a hydration kinetics model, which was based on nucleation and growth of products and diffusion of free water. Then, based on the action mechanism of set-controlling admixtures on cement, we modified the hydration kinetics model. Finally, the temperature development of concrete mixed with set-controlling admixtures was predicted by the new hydration kinetics model.

## 2. Relevant Kinetics Mechanisms and Models

For the hydration kinetics models of cement, Xie et al. [11] summarized in the review article in 2011. They summarized that the kinetics mechanisms of cement included surface reaction, nucleation, growth, space filling, densification and diffusion. The overall hydration process has five main stages in the typical heat-evolution curve. The five main stages are the dissolution period, induction period, acceleration period, deceleration period and slow hydration period. Since the last three periods of cement hydration mainly dominate in the concrete temperature rise, this paper concentrates on studying the acceleration period, deceleration period and slow hydration period. The nucleation and growth of hydration product are deemed to be the driving mechanisms of the acceleration period and the initial part of the deceleration period. Free water diffusion is the driving mechanism of the remaining part of the deceleration and slow hydration periods.

### 2.1. Boundary Nucleation and Growth

It has been well-accepted that the acceleration period and the initial part of the deceleration period are generated due to nucleation and growth (NG) of hydration products, primarily calcium–silicate–hydrate (C–S–H). To accurately describe the hydration kinetics of cement, there are different hypotheses to the nucleation and growth model. 

The Avrami model [16] is a classic homogeneous nucleation model, which assumes that the nucleation of C–S–H randomly takes place on the surface of grains. Subsequently, Cahn [11] suggested a nucleation model for the systems where the nucleation randomly took place on the surface of solids rather than in any available space. Overall, the NG model hypothesizes that the perpendicular impingement explains the transition from acceleration to deceleration with the growing hydration products of neighboring particles. Masoero et al. [17] concluded that the hypothesis of Cahn’s and modified Cahn’s model had a huge flaw, which was that the hypothesis of the NG model did not consider the influence of the water to cement ratio (w/c). Unless the w/c ratio is unusually low, there is enough space for complete hydration. Finally, Masoero et al. [17] proposed a hypothesis to solve the problem: during the early reaction period the C–S–H is restricted to form only within a region close to the cement or C_3_S particles, which we call the “reaction zone’’. However, many researchers [18,19] using direct microstructural observation found the impingement with the growing hydration products at same particles, but the hypothesis proposed by Masoero et al. neglected that.

Therefore, the existing hypotheses about the nucleation and growth model cannot precisely describe the hydration process of cement. New hypotheses need to be proposed to better fit the hydration kinetics process of cement.

### 2.2. Diffusion

To explain the behaviors in the slow hydration period, diffusion of free water is deemed to be its driving mechanism. Many works have applied Fuji and Kondo’s model [20] to describe the diffusion period of cement hydration. The mathematical expression of Fuji and Kondo’s model is given in Equation (1).
(1)$$\alpha =1-{\left[{\left(1-\alpha \right)}^{\frac{1}{3}}-\frac{{\left[2D_{e}\left(t-t^{*}\right)\right]}^{\frac{1}{2}}}{R_{0}}\right]}^{3}$$
where *α* is the degree of hydration, *R*_0_ is radius of original cement grain, *t* is the hydration time, *t** is the time at which diffusion of water becomes the limiting mechanism, *D_e_* is the effective diffusion coefficient of water through the C–S–H and *D_e_* is supposed to be constant.

However, Fuji and Kondo’s model has shortcomings in describing the diffusion period of cement hydration. Firstly, this model assumed that the cement particles were uniform, ignoring the particle size distribution (PSD) effect on the cement hydration. Secondly, this model assumed that the effective diffusion coefficient of water through the C–S–H was constant. However, with the increase of degree of hydration of cement, the porosity and tortuosity of C–S–H was changed, as shown in Figure 1, corresponding to altering the effective diffusion coefficient of water through the C–S–H. Aiming at the above problems, some assumptions of Fuji and Kondo’s model must be reworked.

## 3. Construction of the New Hydration Kinetics Model

### 3.1. Nucleation and Growth Model

To solve the shortcomings of the existing models, two assumptions are given: (1) the impingement of hydration products does not only occur on the adjacent particles but also occurs on the adjacent C–S–H nuclei in same particle and (2) the C–S–H product is restricted to form only within a region close to the cement particles, which is called “reaction zone” and we consider that during the early reaction period the adequate free water is the water in the “reaction zone”. The schematic diagram of the derivation process of the nucleation and growth model is given in Figure 2.

Firstly, it is assumed that the growth of C–S–H has different directions. The tangential growth rate is *G*_1_ and the normal growth rate of *G*_2_, *g* = *G*_1_/*G*_2_. The growth in length of the single ellipsoid nucleate in time *τ* is *G*_2_(*t* − *τ*), then the cross-sectional area *A*(*y*) of the single ellipsoid nucleate at the position of the distance y on the cement surface is:(2)$$A\left(y\right)=\pi g\left(G_{2}{\left(t-\tau \right)}^{2}-y^{2}\right)$$

Then, considering the impingement between the adjacent C–S–H crystal nuclei of the same cement particle, at time *t*, the actual total cross-sectional area *Y*(*y*,*t*) of the position away from *y* is: (3)$$Y\left(y,t\right)=1-\exp\left(-\int_{0}^{t}A\left(y\right)I\left(\tau \right)d\tau \right)$$
where *I*(*τ*) is nucleation rate. Generally, *I*(*τ*) is constant.

Considering the inter-space impingement of different cement particles, the total C–S–H volume *V_CSH_* in the cement hydration space can be obtained as follows:(4)$$V_{CSH}=V_{free}(1-\exp(-\int_{0}^{G_{2}t}a_{BV}Y\left(y,t\right)dy))$$
where *a_BV_* is the area of the phase boundary per unit volume, which can expressed by:(5)$$a_{BV}=\frac{SSA}{V_{free}}$$
where *SSA* is the specific surface area of cement and *V_free_* is the volume of effective free water. Here, we consider the free water is the water within “reaction zone”, so the volume of free water can be calculated by:(6)$$V_{free}=\int_{0}^{\infty }w\left(R_{0}\right)\frac{4}{3}\pi \left(R_{lim}^{3}-R_{0}^{3}\right)dR_{0}$$
where *w*(*R*_0_) is the distribution function of cement, *R_lim_* is the radius of “reaction zone” and *R*_0_ is the initial radius of cement particle.

The distribution function of cement particles adopts the Rosin–Rammler distribution function, which can be expressed by:(7)$$w\left(R_{0}\right)=\frac{mR_{0}^{m-1}}{R_{50}^{m}}\exp\left(-{\left(\frac{R_{0}}{R_{50}}\right)}^{m}\right)$$
where *R*_50_ is median particle radius of cement and *m* is constant.

For the “reaction zone” of cement hydration, Tulio et al. [21] consider that the “reaction zone” is related to initial radius of cement particle and water to cement. The radius of the “reaction zone ” *R_lim_* can be expressed as:(8)$$R_{lim}=R_{0}\sqrt[{3}]{1+\frac{w}{c}\frac{\rho_{c}}{\rho_{w}}}$$
where *w*/*c* is ratio of water to cement, *ρ_c_* is the density of cement and *ρ_w_* is the density of water.

The hydration of C_3_S is assumed to obey the following stoichiometric balance between C_3_S, water, C–S–H and CH [17]:(9)$$C_{3}S+3.1H\to C_{1.7}{\mathrm{SH}}_{1.8}+1.3\mathrm{CH}$$

The density and molar volumes of the C_3_S, water, C–S–H and CH in Equation (9) refer to the Ref. [17]. The densities of four phases are 3.15 g/cm^3^, 1.00 g/cm^3^, 2.05 g/cm^3^ and 2.24 g/cm^3^, respectively. The molar volumes of four phases are 72.5 cm^3^/mol, 18.1 cm^3^/mol, 110.1 cm^3^/mol and 33.1 cm^3^/mol, respectively.

The degree of hydration can be calculated according to:(10)$$\alpha \left(t\right)=\frac{M_{C_{3}S}V_{free}}{m_{C_{3}S}V_{MCSH}}(1-\exp\left(-\int_{0}^{G_{2}t}\frac{SSA}{V_{free}}\left(1-\exp\left(-\int_{0}^{t}\pi gI\left(G_{2}{\left(t-\tau \right)}^{2}-y^{2}\right)d\tau \right)\right)dy)\right)$$
where *M_C3S_* is molar mass of C_3_S, cm^3^/mol, *m_C3S_* is the original mass of C_3_S, g and *V_MCSH_* is molar volume of C–S–H.

### 3.2. Modified Diffusion Model

#### 3.2.1. Effects of the PSD on Hydration Kinetics

In order to consider the effects of particle size distribution (PSD) on the cement hydration, the degree of hydration for an individual cement particle is *α_i_*. The overall degree of hydration *α_t_* of the system can be expressed by:(11)$$\alpha_{t}=\int_{0}^{\infty }w\left(R_{0}\right)\alpha_{i}dR_{0}$$

#### 3.2.2. Effects of the Effective Diffusion Coefficient on Hydration Kinetics

The deposition of C–S–H during the diffusion period of cement hydration will block the pores of C–S–H. Thereby the effective diffusion coefficient of water through the C–S–H changes with the degree of hydration of cement. Park et al. [22,23] considered that the relationship between the effective diffusion coefficient and degree of hydration could be expressed as Equation (3).
(12)$$D_{e}=k_{\varphi }D_{e0}$$
where *D_e_*_0_ is the initial effective diffusion coefficient of water through the C–S–H. *k_φ_* is the pore-structure-affecting factor.

To obtain the pore-structure-affecting factor, two assumptions are given: (1) the sedimentation of hydration products on the pores mainly occurs on the pore wall and is evenly deposited layer by layer and (2) the pore of unhydrated cement particle is regarded as a large pore perpendicular to the diffusion direction, which is shown in Figure 1.

Based on above assumption, *k_φ_* can be expressed as:(13)$$k_{\varphi }=\frac{\varphi_{0}}{\xi }{\left(\frac{d\left(x,t\right)}{d_{0}}\right)}^{2}$$
where *ξ* is the deflection of the pore, *φ*_0_ is the initial porosity, *d*_0_ is the initial pore diameter and *d*(*x*,*t*) is the pore diameter corresponding to hydration time *t*, which is related to the production of C–S–H. The change of pore diameter Δ*d*(*x*,*t*) can be expressed as:(14)$$\Delta d\left(x,t\right)=d_{0}-d\left(x,t\right)=k_{d}\left(C_{csh}-C_{csh}^{0}\right)d_{0}$$
where *k_d_* is precipitation constant of C–S–H, *C_csh_* is the concentration of C–S–H and *C*^0^*_csh_* is the initial concentration of C–S–H.

Equation (14) can be converted to Equation (15):(15)$$d\left(x,t\right)=d_{0}-k_{d}\left(C_{csh}-C_{csh}^{0}\right)d_{0}$$

Based on Equation (15), Equation (13) can be converted to Equation (16):(16)$$k_{\varphi }=\frac{\varphi_{0}}{\xi }{\left(1-k_{d}\left(C_{csh}-C_{csh}^{0}\right)\right)}^{2}$$

Equation (12) can be converted to Equation (17):(17)$$D_{e}=\frac{\varphi_{0}}{\tau }{\left(1-k_{d}\left(C_{csh}-C_{csh}^{0}\right)\right)}^{2}D_{e0}$$

The modified Fuji and Kondo’s model can be expressed as:(18)$$\alpha^{j}=\int_{0}^{\infty }w\left(R_{0}\right)(1-\left({\left(1-\alpha^{j}\right)}^{\frac{1}{3}}-\frac{{\left(2\frac{\varphi_{0}}{\xi }{\left(1-k_{d}\left(C_{csh}-C_{csh}^{0}\right)\right)}^{2}D_{e0}\left(t-t^{*}\right)\right)}^{2}}{R_{0}}\right)dR_{0}$$

#### 3.2.3. Effects of Temperature on Hydration Kinetics

As is well known, temperature affects the hydration of cement, which can be analyzed using Arrhenius’s law as follows:(19)$$I=I_{293K}\exp\left(-\frac{E}{R}\left(\frac{1}{T}-\frac{1}{293}\right)\right)$$
(20)$$G=G_{293K}\exp\left(-\frac{E}{R}\left(\frac{1}{T}-\frac{1}{293}\right)\right)$$
(21)$$D_{e}=D_{e293K}\exp\left(-\frac{E}{R}\left(\frac{1}{T}-\frac{1}{293}\right)\right)$$
where *I_293K_*, *G_293K_* and *D_e293K_* are the values of *I_293K_*, *G_293K_* and *D_e293K_* at 293K, respectively; *E* is the apparent activation energy, kJ/mol; *R* is the gas constant, which is 8.314 J/(mol·K) and *T* is the curing temperature for hydration of cement, K.

#### 3.2.4. Effects of Cement Composition on Hydration Kinetics

It is assumed that the hydration of C_3_S plays a main role in the nucleation and growth period. However, each mineral component of cement clinker contributes to the hydration of cement in the diffusion period. The total degree of cement hydration *α* can be calculated as follows:(22)$$\alpha =\overset{4}{\underset{j=1}{\sum }}\alpha^{j}p_{j}$$
where *α^j^* is the degree of each mineral component of cement clinker hydration, *j* ∈ {C_3_S, C_2_S, C_3_A, C_4_AF} and *p_j_* is the mass fraction of each mineral components of cement clinker.

#### 3.2.5. Determination of the Transition Time

Ouzia et al. [18] indicated that the inner CSH starts forming at about 20 hand then the mechanism turns into the diffusion period. Zhang et al. [19] have carried out a significant amount of analysis and consider that when the area fraction of the surface covered by outer CSH reaches 99%, the hydration mechanism transforms from nucleation and growth-controlling to diffusion-controlling. In this paper, we assume that the transition in mechanism occurs while *V** equals to 99%. Therefore, we can use the following equation solving *t**:(23)$$V\left(t^{*}\right)=1-\exp(-\int_{0}^{G_{c}t^{*}}\frac{SSA_{c}}{V_{free}}\left(1-\exp\left(-\int_{0}^{t^{*}}\pi gI_{c}\left(G_{c}{\left(t^{*}-\tau \right)}^{2}-y^{2}\right)d\tau \right)\right)dy)$$

Based on the above analysis, the hydration kinetics models of cement can be expressed as following equation.
(24)$$\frac{d\alpha \left(t\right)}{dt}=\left\{ \begin{array}{c} \frac{M_{C_{3}S}V_{free}}{m_{C_{3}S}V_{MCSH}}(1-\exp\left(-\int_{0}^{G_{2}t}\frac{SSA}{V_{free}}\left(1-\exp\left(-\int_{0}^{t}\pi gI\left(G_{2}{\left(t-\tau \right)}^{2}-y^{2}\right)d\tau \right)\right)dy)\right)\;t\leq t^{*}\\ \overset{4}{\underset{j=1}{\sum }}p_{j}\frac{d(\int_{0}^{\infty }w\left(R_{0}\right)\left(1-\left({\left(1-\alpha^{j}\right)}^{\frac{1}{3}}-\frac{{\left(2\frac{\varphi_{0}}{\xi }{\left(1-k_{d}\left(C_{csh}-C_{csh}^{0}\right)\right)}^{2}D_{e0}\left(t-t^{*}\right)\right)}^{2}}{R_{0}}\right)dR_{0}\right)}{dt}\;t>t^{*} \end{array}\right.$$

### 3.3. Hydration Kinetics Model for Cement with Set-Controlling Admixture

#### 3.3.1. Influence of Retarder on Cement Hydration

Research carried out over the past 50 years on the mechanisms of action of retarders is summarized in [24,25,26]. There are basically four regimes: (1) calcium complexation preventing precipitation of portlandite, (2) formation of a semipermeable layer preventing hydration, (3) adsorption of retarder on the anhydrous surface and (4) nucleation and growth-poisoning of hydrates. However, this research has found that the action mechanism of retarder seems unlikely to be one of above four regimes. For example, very strong chelators can be moderate retarders while strong retarders can be moderate chelators [27]. In addition, using SEM observation during the induction period, the semipermeable layer cannot be found except in rare hydration products [28].

Yan et al. [29] researched the influence of a temperature-rise inhibitor (TRI) on the hydration of cement, and they found that a TRI not only affected the nucleation and growth of C–S–H but also absorbed onto the cement surface, thereby blocking the surface. Perez et al. [30] used conductimetry and solution analysis to investigate the sodium gluconate retardation to cement, and they concluded that the cement surface formed a adsorbed gluconate layer, approximately 3 nm in thickness. Zajac et al. [31] used quantitative XRD to investigate the influence of retarders (for example, gluconate and tartrate) on the hydration products, and they found that the retarders inhibited the formation of AFt and hydrogarnet.

Based on above analysis, the influence of retarders on the nucleation and growth of hydration products shows in two ways: (1) the retarder occupies the nucleus of hydration products, thereby reducing the rate of nucleation and rate of growth and (2) the retarder blocks the surface of cement, reducing the area of boundary of nucleation and growth.

#### 3.3.2. Influence of Accelerator on Cement Hydration

Due to the variety of accelerators, their mechanism of action is also different. Cheung et al. [24] have summarized the mechanism of action of accelerators. They concluded that a model attempting to describe the interaction of accelerators with cement, must separately deal with: (1) acceleration of pitting on the bare silicate surface during the induction period; (2) acceleration of CH nucleation, which may control saturation levels and affect etch pitting; (3) acceleration of C–S–H nucleation, which affects the reaction rate after the end of induction period and (4) acceleration of C–S–H growth after nucleation. We describe the hydration process after the end of induction period, so the third and fourth questions must be dealt with.

Except for the questions proposed by Cheung et al., many researchers [32,33] found that the dosage of organic accelerators (for example, triethanolamine and triisopropaniamine) over a certain amount inhibits the hydration of cement. Lu et al. [34] studied the influence of triethanolamine on the hydration of cement, and they found that the C_3_S surface formed a complex layer which inhibited the dissolution of C_3_S and precipitation of C–S–H during and after the induction period. This research shows that organic accelerators can reduce the area of boundary of nucleation and growth.

Based on the above analysis, the influence of accelerators on the nucleation and growth of hydration products occurs in two ways: (1) accelerators increase the rate of nucleation and rate of growth and (2) organic accelerators can block the surface of cement, reducing the area of boundary of nucleation and growth.

#### 3.3.3. Modified Hydration Kinetics Model

The incorporation of set-controlling admixture alters the rate of nucleation and growth. This paper introduces the nucleation rate effect coefficient Ieff and the growth-rate-effect coefficient *G_eff_*. The rate of nucleation *I_s_* and rate of growth *G_s_* under the action of set-controlling admixture can be expressed as:(25)$$I_{s}=I_{eff}\times I$$
(26)$$G_{s}=G_{eff}\times G$$

The organic set-controlling can adsorb on the surface of cement, reducing the area of the boundary of nucleation and growth. This adsorption property reduces the specific surface area. Therefore, the specific surface area reduction coefficient *SSA_r_* is introduced. The specific surface area under the action of set-controlling admixture *SSA_s_* can be expressed as:(27)$$SSA=SSA_{r}\times SSA$$

Finally, the nucleation and growth model of cement under the action of set-controlling admixture can be expressed as:(28)$$\frac{d\alpha \left(t\right)}{dt}=\left\{ \begin{array}{c} \frac{M_{C_{3}S}V_{free}}{m_{C_{3}S}V_{MCSH}}\frac{d(1-\exp\left(-\int_{0}^{G_{c}t}\frac{SSA_{c}}{V_{free}}\left(1-\exp\left(-\int_{0}^{t}\pi gI_{c}\left(G_{c}{\left(t-\tau \right)}^{2}-y^{2}\right)d\tau \right)\right)dy)\right)}{dt}\;t\leq t^{*}\\ \overset{4}{\underset{j=1}{\sum }}p_{j}\frac{d(\int_{0}^{\infty }w\left(R_{0}\right)\left(1-\left({\left(1-\alpha^{j}\right)}^{\frac{1}{3}}-\frac{{\left(2\frac{\varphi_{0}}{\xi }{\left(1-k_{d}\left(C_{csh}-C_{csh}^{0}\right)\right)}^{2}D_{e0}\left(t-t^{*}\right)\right)}^{2}}{R_{0}}\right)dR_{0}\right)}{dt}\;t>t^{*} \end{array}\right.$$

## 4. Predicting Temperature Development of Concrete with Set-Controlling Admixture

### 4.1. Materials and Methods

#### 4.1.1. Materials

The cement was P.O 42.5 (Chinese standard GB175-2007) ordinary Portland cement (OPC). The detailed oxide compositions of the OPC are shown in Table 1. Citric acid (CA), sodium gluconate (SG), triethanolamine (TEA) and calcium chloride (CC) were chosen as set-controlling admixtures, and their purity was analytically pure.

#### 4.1.2. Methods

The isothermal calorimetry was conducted by an 8-channel TAM Air calorimeter (TA Instruments, New Castle, USA). 10.0 g of the prepared samples with water to binder ratio of 0.5 was used. Isothermal calorimetry of samples was performed at 20 °C for 48 h. The composition of mixes is shown in Table 2.

A small slice about 2 mm thick was taken from the cement paste. The small slice was examined by scanning electron microscope (SEM, FEI Quanta 200, FEI Co., Hillsboro, OR, USA) with an accelerating voltage of 15 kV.

An adiabatic curing test method measures the temperature rise of 50 L of freshly cast concrete. The schematic diagram of an adiabatic temperature rise of concrete testing device are shown in Figure 3. Proportions of concrete mix are given in Table 3.

### 4.2. Analysis of Hydration Kinetic

The results of fitting using Equation (28) to the calorimetric data of cement with different set-controlling admixtures are shown in Figure 4. The value of hydration kinetics parameters of cement with set-controlling admixture is given in Table 4.

As shown in Table 4, after the cement blending, the retarder, for example, citric acid or sodium gluconate, would reduce the nucleation rate and growth rate of hydration products. Generally, the retarder can complex the Ca^2+^ to alter the concentration of Ca^2+^, thereby inhibiting the nucleation of C–S–H. It must be realized that the retarder can adsorb on the surface of cement particles, and reduce the area of boundary of nucleation and growth, thereby reducing the precipitation of C–S–H on the surface. In the diffusion period, the retarder reduces the number of hydration products (Figure 5), and less deposition of hydration products will block the pores, thereby enhancing free water diffusion.

In addition, after the cement blending the accelerator, for example, triethanolamine or calcium chloride, would increase the nucleation rate and growth rate of hydration products. The triethanolamine can promote the dissolution of C_3_A and C_3_S, increasing the concentration of Ca^2+^ and promoting the nucleation and growth of hydration products. The calcium chloride can flocculate hydrophilically, producing a more permeable C–S–H layer around the cement grains at the early hydration stage, promoting the nucleation and growth of hydration products. However, triethanolamine can adsorb on the surface of cement particles, so the addition of triethanolamine must be controlled, or triethanolamine will inhibit the hydration of cement. In the diffusion period, the accelerator increases the number of hydration products (Figure 5), and the greater deposition of hydration products will block the pores, thereby reducing the diffusion of free water.

The relationships between apparent activation energy and nucleation rate (*I_s_*), growth rate (*G_s_*) and effective diffusion coefficient (*D_e_*) are shown in Figure 6. The following rules can be summarized: (1) the nucleation rate increases with the increase of apparent activation energy, and this is a linear relationship; (2) the growth rate increases with the increase of apparent activation energy, and this is an exponential relationship and (3) the effective diffusion coefficient decreases with the increasing of apparent activation energy, and this is a linear relationship.

### 4.3. Predicting the Adiabatic Temperature Rise History of Concrete

The temperature rise of concrete is caused by the heat release from cement; this increases with the increase of hydration degree. The heat release rate from concrete is related to the mass, composition and hydration rate of cement. The rate of heat release from concrete can be expressed as:(29)$$\frac{dQ}{dt}=m_{c}H_{u}\frac{d\alpha }{dt}$$
where *dQ*/*dt* is rate of heat release from concrete, *m_c_* is the mass of cement in concrete, *H_u_* is the ultimate heat release of cement and *dα*/*dt* is the hydration rate of cement.

The ultimate heat release of cement can be calculated by the following empirical formula [35]:(30)$$H_{u}=500p_{C_{3}S}+260p_{C_{2}S}+866p_{C_{3}A}+420p_{C_{4}AF}+624p_{SO_{3}}+1186p_{f-CaO}+850p_{MgO}$$
where *p_C3S_*, *p_C2S_*, *p_C3A_*, *p_C4AF_*, *p_SO3_*, *p_f-CaO_* and *p_MgO_* are the mass fraction of C_3_S, C_2_S, C_3_A, C_4_AF, SO_3_, *f*-CaO and MgO, respectively.

The heat capacity of concrete is related to the heat capacity of its component. The heat capacity of concrete is calculated by the weighted average of heat capacity of each component, which can be expressed as:(31)$$C\left(t\right)=C_{c}p_{c}+C_{a}p_{a}+C_{w}\left(1-\alpha \right)p_{w}$$
where *C_c_*, *C_a_* and *C_w_* are the heat capacity of cement, aggregate and water, respectively. *p_c_*, *p_a_* and *p_w_* are the mass fraction of cement, aggregate and water, respectively.

The increment temperature rise of concrete can be expressed as:(32)$$\Delta T=\frac{dQ\left(t\right)}{C\left(t\right)}$$

The adiabatic temperature rise of concrete can be expressed as:(33)$$T=T+\frac{dQ\left(t\right)}{C\left(t\right)}$$

The flow chart of numerical simulation process is shown in Figure 7. The adiabatic temperature is calculated by Figure 7, and the predicted results is shown in Figure 8.

As shown in Figure 8, the modeled results generally agree with the experimental results. The correlation coefficients and the standard deviation between the modeled results and experimental results for adiabatic temperature rise tests are approximately 0.97 and 2 °C. However, in some cases, such as the concrete with a large amount of calcium chloride, the modeled result shows disagreement with the experimental result. This difference comes from the fact that a large amount of calcium chloride does not evenly act on the cement in the concrete, resulting in the modeled result disagreeing with experimental result.

## 5. Conclusions

Firstly, based on analysis of the mechanism of cement hydration, a new hydration kinetics model of cement was constructed. Secondly, the hydration kinetics model was used to investigate the influence of set-controlling admixture on cement hydration. Finally, a model of adiabatic temperature rise of concrete was constructed to predict the temperature development of concrete. Based on the model and analysis, the following conclusions are obtained.
The influence of retarders on the nucleation and growth of hydration products shows in two ways: (1) the retarder occupies the nucleus of hydration products, thereby reducing the rate of nucleation and rate of growth and (2) the retarder blocks the surface of cement, reducing the area of boundary of nucleation and growth.The influence of accelerators on the nucleation and growth of hydration products shows in two ways: (1) the accelerators increase the rate of nucleation and rate of growth and (2) organic accelerators can block the surface of cement, reducing the area of boundary of nucleation and growth.The nucleation rate increases with the increase of apparent activation energy, and this is a linear relationship. The growth rate increases with the increasing of apparent activation energy, and this is an exponential relationship. The effective diffusion coefficient decreases with the increasing of apparent activation energy, and this is a linear relationship.The hydration rate of cement determines the heat release rate of concrete. Furthermore, the temperature development of concrete can be calculated by the hydration degree of cement and the modeled results agree with the experimental results.

Although we have tried our best to improve the model, there are still some problems to be solved. Firstly, there are too many assumptions in the new hydrtion kinetics model. Our later research will verify our hypothesis through experiments, which will make the new model construction more convincing; secondly, based on the analysis of the mechanism of cement hydration, a new hydration kinetics model of cement was constructed, but merely to predict the temperature development of concrete. Our later research will focus on the prediction of the compressive strength, durability, porosity, etc., of concrete by the new model and check the validity of the model.

## Figures and Tables

**Figure 1 materials-16-00497-f001:**
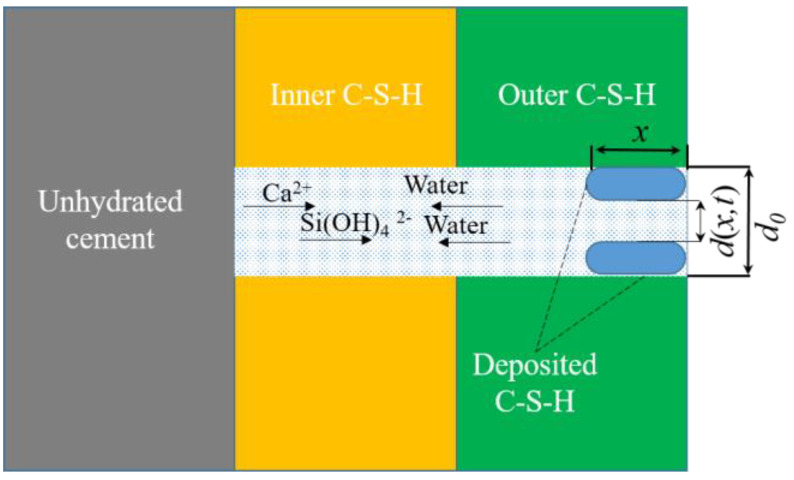
The deposition of CSH during the diffusion period of cement hydration.

**Figure 2 materials-16-00497-f002:**
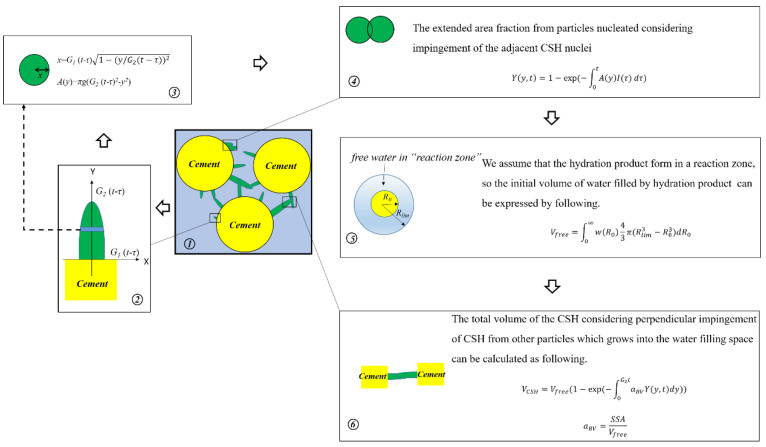
The schematic diagram of derivation process of nucleation and growth model.

**Figure 3 materials-16-00497-f003:**
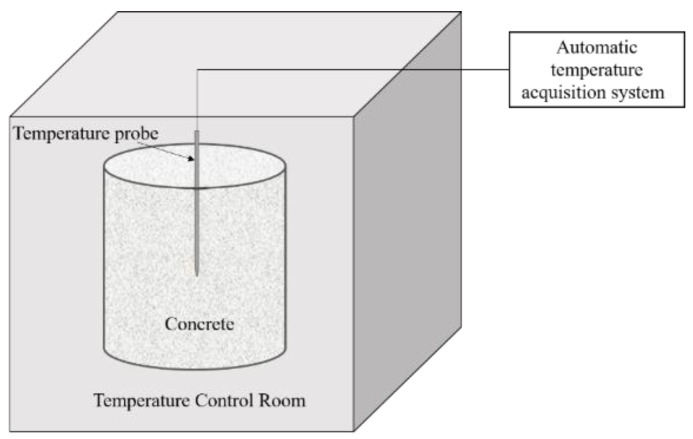
Schematic diagram of adiabatic temperature rise of concrete testing device.

**Figure 4 materials-16-00497-f004:**
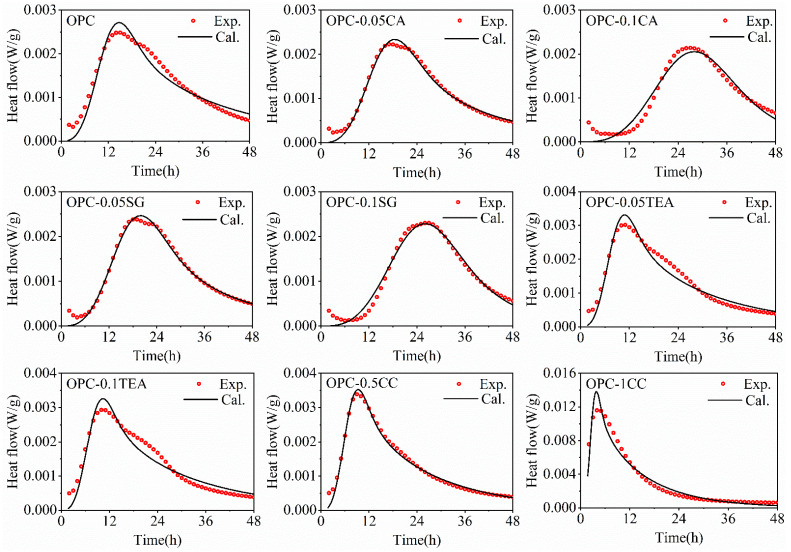
The results of fitting using Equation (28) to the calorimetric data of cement with different set-controlling admixtures.

**Figure 5 materials-16-00497-f005:**
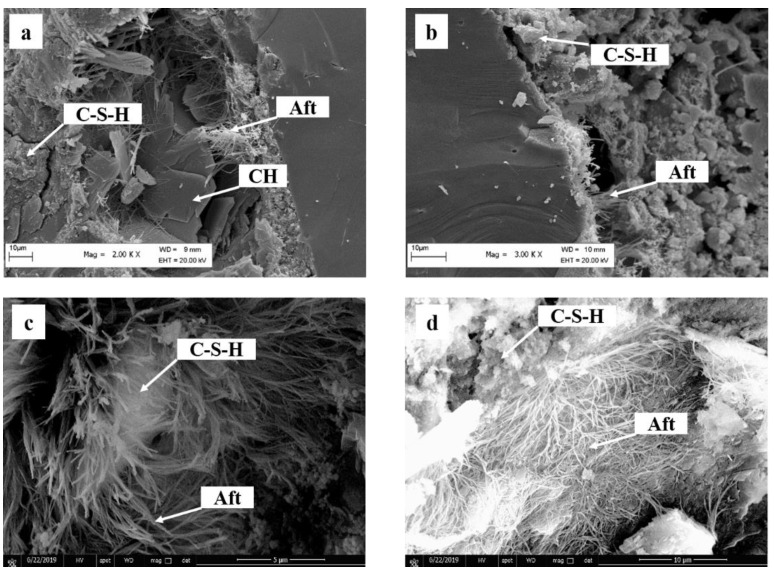
Microstructure of (**a**) cement, (**b**) cement with CA, (**c**) cement with TEA and (**d**) cement with CC.

**Figure 6 materials-16-00497-f006:**
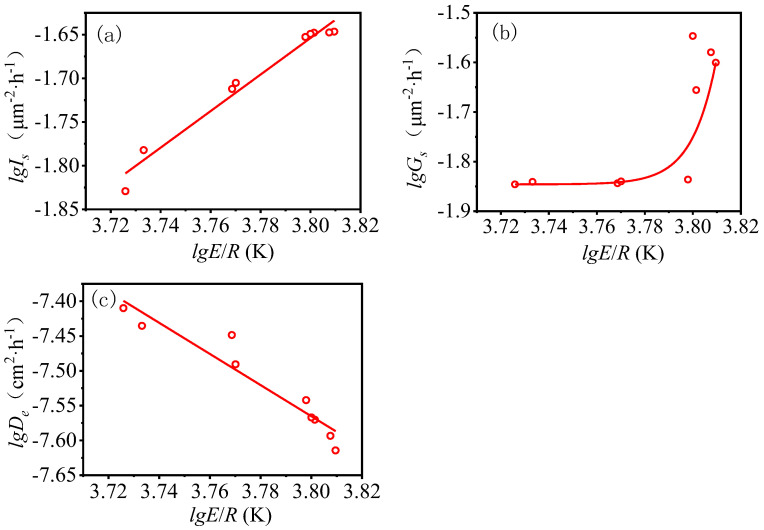
The relationship of hydration kinetics parameters between apparent activation energy and (**a**) *I_s_*, (**b**) *G_s_* and (**c**) *D_e_*, respectively.

**Figure 7 materials-16-00497-f007:**
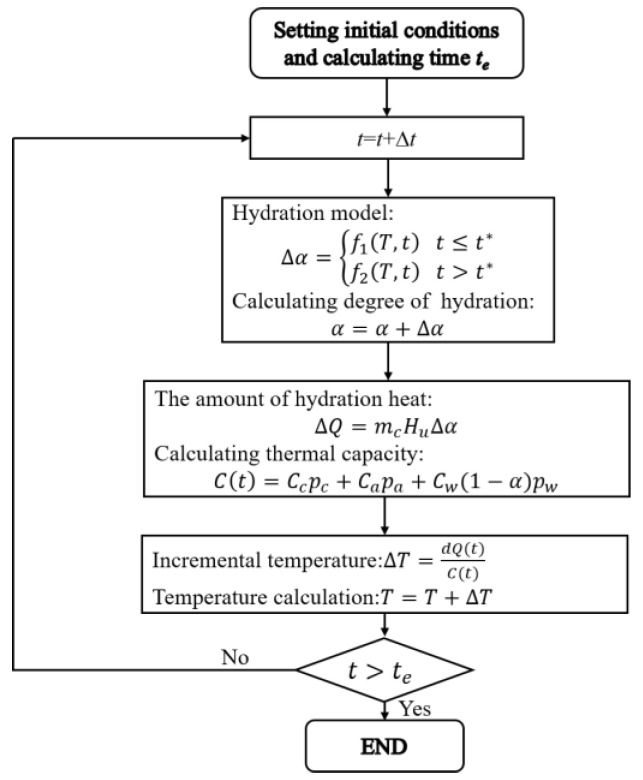
The flow chart of numerical simulation process.

**Figure 8 materials-16-00497-f008:**
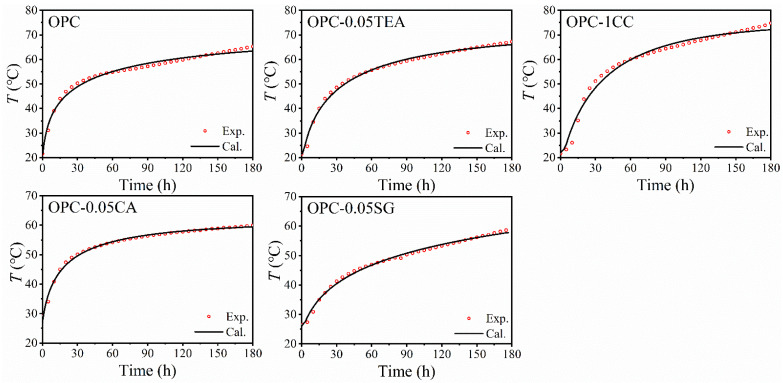
The predicted results of adiabatic temperature rise of concrete with different set-controlling admixtures.

**Table 1 materials-16-00497-t001:** The oxide compositions of the cement (%).

SiO_2_	Al_2_O_3_	Fe_2_O_3_	CaO	MgO	K_2_O	Na_2_O	SO_3_	Ignition Loss
20.71	5.29	3.55	62.81	1.43	1.1	0.167	1.54	1.36

**Table 2 materials-16-00497-t002:** Composition of mixes (wt.%).

Sample Code	OPC	CA	SG	TEA	CC
OPC	100	-	-	-	-
OPC-0.05CA	100	0.05	-	-	-
OPC-0.1CA	100	0.1	-	-	-
OPC-0.05SG	100	-	0.05	-	-
OPC-0.1SG	100	-	0.1	-	-
OPC-0.05TEA	100	-	-	0.05	-
OPC-0.1TEA	100	-	-	0.1	-
OPC-0.5CC	100	-	-	-	0.5
OPC-1CC	100	-	-	-	1

**Table 3 materials-16-00497-t003:** Proportions of concrete mix (kg/m^3^).

Sample Code	Water	OPC	Fine Aggregate	Coarse Aggregate	CA	SG	TEA	CC
PC	214.5	390	825	1238	-	-	-	-
PC-0.05CA	214.5	390	825	1238	0.195	-	-	-
PC-0.05SG	214.5	390	825	1238	-	0.195	-	-
PC-0.05TEA	214.5	390	825	1238	-	-	0.195	-
PC-1CC	214.5	390	825	1238	-	-	-	0.390

**Table 4 materials-16-00497-t004:** Hydration kinetics parameters of cement with set-controlling admixture.

Sample Code	*I_s_* (μm^−2^·h^−1^)	*I_eff_*	*G_s_* (μm^−2^·h^−1^)	*G_eff_*	*SSA_s_* (m^2^·kg^−1^)	*SSA_r_*	*D_e_* (cm^2^·h^−1^)	E/R (K)
OPC	0.022250	1.000	0.014580	1.000	369.5	1.000	2.87 × 10^−8^	6280
OPC-0.05CA	0.019402	0.872	0.014332	0.983	341.0	0.923	3.56 × 10^−8^	5870
OPC-0.1CA	0.014819	0.666	0.014259	0.978	334.0	0.904	3.89 × 10^−8^	5320
OPC-0.05SG	0.019714	0.886	0.014449	0.991	345.1	0.934	3.23 × 10^−8^	5890
OPC-0.1SG	0.016510	0.742	0.014420	0.989	335.9	0.909	3.67 × 10^−8^	5410
OPC-0.05TEA	0.022495	1.011	0.022089	1.515	359.9	0.974	2.69 × 10^−8^	6330
OPC-0.1TEA	0.022562	1.014	0.025092	1.721	354.4	0.959	2.43 × 10^−8^	6450
OPC-0.5CC	0.022428	1.008	0.032384	2.907	369.5	1.000	2.71 × 10^−8^	6310
OPC-1CC	0.022517	1.012	0.030316	3.451	369.5	1.000	2.55 × 10^−8^	6420

## Data Availability

Not applicable.

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
