# Peer review of "Prediction of Temperature Development of Concrete with Set-Controlling Admixture Based on a New Hydration Kinetics Model"

_materials, 2023, doi:10.3390/ma16020497_

Round 1

Reviewer 1 Report

Since the similarity index, plagiarism is 34%, I recommend correcting the work.

I recommend completing the abstract with importance, the necessity that led to the realization of this work.

For the Introduction section, the authors studied few documentary sources. This section could be developed.

Zheng, J.J.; Zhang, J.; Scherer, G. Prediction of the degree of hydration at initial setting time of cement paste with particle agglomeration, Cement and Concrete Research 42, 2012, 1280–1285.

X. Pang et al. Cement hydration kinetics study in the temperature range from 15 â—¦C to 95 â—¦C, Cement and Concrete Research 148 (2021) 106552

Section 3.2.4. Effects of cement composition on hydration kinetics

In the experimental work, did you also analyze Influences of w/c ratio on cement hydration kinetics?

The references also have numbers, they also have the numbers in square brackets. References should be written according to the writing instructions of the journal, and the completion of the references with current, recent sources,

Considering that the hydration degree of cement is an important property, the higher this value is, the more cement particles will be transformed into hardened cement. This paper establishes a hydration kinetics model taking into account the effect of water on cement, fineness, particle size distribution, cement composition and temperature and predicts the evolution of concrete temperature, I congratulate the authors for the obtained experimental results.

I recommend publishing the work after minor corrections.

The subject is original and relevant in this field.

The conclusions are consistent with the evidence and arguments presented.

Reviewer 2 Report

The authors conducted a theoretical study using concrete with set-controlling admixture based on a new hydration kinetics model, however, there are major issues that must be solved before publication.

Firstly, based on the analysis of the mechanism of cement hydration, a new hydration kinetics model of cement was constructed to investigate the influence of set-controlling admixture on cement hydration and to predict the temperature development of concrete, what about the compressive strength and durability of the cementitious product? What about porosity with temperature development? What about the density of samples?

Secondly, I recommend more clearly emphasizing this study's novelty in the introduction and conclusion to show the contribution of the authors in this field obtained in this study.

Thirdly, I found the manuscript to be rather difficult to read. I recommend improving the style of this manuscript, especially in the methodology and results.

Finally, I believe that the theme of this manuscript can be consistent with the theme of materials. At the same time, the manuscript needs to be improved. Authors should edit the manuscript by the guidelines of the journal.

Reviewer 3 Report

In my opinion, this article is very interesting. 

The authors carry out an adequate study, related to the kinetics of the heat of hydration in concrete. The only change that I propose is to improve the conclusions section, including a general conclusion of the work and indicating future lines of investigation.

Author Response

According your suggestion, we have revised the conclusion section and have presented the future research, please refer to the revised manuscript.

Reviewer 4 Report

This manuscript evaluates the “Prediction of temperature development of concrete with  set-controlling admixture based on a new hydration kinetics  model”. The manuscript is elaborately described and contextualized with the help of previous and present theoretical background. All the references cited are relevant to this area of research. The methods/analytical study are clearly stated. The result and discussion section are clearly presented. The manuscript needs minor revision and require the following modifications before the acceptance.

1. Abstract: Mention your research need.

2. Present your research recommendation in the abstract.

3. Remove the key words ‘model’

4. What is the novelty of your research?

5. Fig 5. Mark the features in the SEM images and then discuss it.

6. Include your research recommendation for future research in the conclusion section

7. Use more recent relevant works like 2021,2022 for citing your works.

Round 2

Reviewer 2 Report

Accepted in current form.